# Dutasteride treatment and its effect on standardized uptake values in prostate-specific membrane antigen-PET imaging: A pilot study

Lucas Praetzel[1], Irene A. Burger[2,3,4], Jan H. Rüschoff[5], Niels J. Rupp[5], Daniel Eberli[1], Benedikt Kranzbühler[1]*

1 Department of Urology, University Hospital of Zurich, Zürich, Switzerland, 2 Department of Nuclear Medicine, University Hospital of Zurich, Zürich, Switzerland, 3 Department of Nuclear Medicine, Cantonal Hospital Baden, University of Zurich, Baden, Switzerland, 4 Department of Health Sciences and Technology, Federal Institute of Technology Zurich, Zurich, Switzerland, 5 Department of Pathology, University Hospital of Zurich, Zürich, Switzerland

* b.kranzbuehler@me.com

## Abstract

### Background

Prostate-specific membrane antigen (PSMA)-based imaging has become an increasingly important diagnostic tool in prostate cancer, though limited by low surface expression of PSMA in some patients. Previous studies have demonstrated that dutasteride can induce PSMA expression *in vitro* and *in vivo*. This pilot study aimed to evaluate the impact of short-term dutasteride treatment on standardized uptake values (SUVmax) in PSMA PET imaging and the immunohistochemical expression of PSMA for the first time in humans.

### Methods

Four prostate cancer (PCa) patients underwent an initial PSMA PET/MRI of the prostate. Afterwards, all patients received 0.5 mg of oral dutasteride once daily for seven days. Subsequently, a second PSMA PET/MRI of the prostate and a template biopsy were performed. We compared the maximum standardized uptake value (SUVmax) of PSMA-positive lesions before and after dutasteride treatment. Additionally, histopathological specimens from PSMA-positive lesions and negative controls were analyzed for Gleason score and PSMA expression.

### Results

An increase in SUVmax was observed in all patients following short-term dutasteride treatment. Histological analysis confirmed prostate cancer with an ISUP grade of ≥ 2 in PSMA-positive lesions that exhibited increased SUVmax following short-term stimulation. One PSMA-positive lesion, which showed a decrease in SUVmax after stimulation, was negative for prostate cancer on biopsy.

**Data availability statement:** All relevant data are within the manuscript.

**Funding:** The author(s) received no specific funding for this work.

**Competing interests:** There are no competing interests.

## Conclusion

This pilot study demonstrated an increase in SUVmax in PSMA-positive prostate cancer lesions following a short-term seven-day course of dutasteride. Short-term dutasteride treatment prior to PSMA-PET imaging may have the potential to enhance detection rates in patients with prostate cancer. Further studies are needed to investigate this effect in larger patient populations.

## Introduction

The prostate-specific membrane antigen (PSMA) was first described and cloned in 1993 by Israeli et al. [1]. This type II transmembrane glycoprotein is overexpressed in up to 90% of PCa cells [2]. It is well-established that higher Gleason scores correlate with increased PSMA expression in PCa cells [3,4]. Even though modern PSMA PET/CTs have a superior diagnostic accuracy than conventional imaging [5], the detection rate remains limited in patients with low-volume disease (low-grade cancer, low PSA) at the time of recurrence [6–8]. Conversely, salvage radiotherapy is most effective in patients with PSA levels below 0.2 ng/mL [9–11]. A negative PSMA PET should therefore not delay a salvage radiotherapy [12].

Several studies have demonstrated increased PSMA expression following stimulation with various compounds. Upregulated PSMA expression has been observed following treatment with Apalutamide, Abiraterone, and Enzalutamide *in vitro* [13,14]. This upregulation must be taken into consideration as up to thirty percent of patients having a PSMA PET for restaging are treated with some form of ADT in metastatic hormonsensitive and castration-resistent prostate cancer. The relationship between PSMA expression and ADT may explain the different detection rates in PSMA PET imaging [15]. Due to these potential side effects [16] of conventional ADT and androgen receptor pathway inhibitors, recent studies have investigated the impact of dutasteride on PSMA expression *in vitro* and *in vivo* [17,18].

This pilot study aimed to evaluate the impact of short-term dutasteride treatment on the detection rate of PSMA imaging (PSMA PET/MRI) in humans. We assessed changes in SUVmax and analyzed histological specimens for International Society of Urological Pathology (ISUP) grade and PSMA expression following dutasteride treatment.

## Materials and methods

### Patients

Overall, four patients were enrolled in this pilot study. These patients were participants in a prospective trial investigating the long-term outcomes of high-intensity focused ultrasound (HIFU) as a focal therapy for PCa [19]. All patients were diagnosed with PCa via prostate biopsy and were discussed at the interdisciplinary tumor board of the University Hospital of Zurich. The board includes specialists from urology, medical oncology, radiation oncology, radiology, pathology, and nuclear medicine, with additional input from other surgical disciplines and supportive care

teams when required. Diagnostic imaging, histopathological findings, and clinical data are reviewed collectively to formulate consensus-based treatment recommendations. Following detailed information, the patients consented to focal HIFU treatment and participation in this ancillary study. The local ethics committee approved the study protocol, and all patients provided written informed consent (BASEC PB_2016–02563, Amendment dated 10.01.2018). Documented patient characteristics included age, PSA value at the time of the first scan, MRI-PI-RADS lesion, initial tumor stage, and ISUP grade group. For comparison, the stated PSA values were the last measurements before dutasteride treatment. This information was obtained from patient records.

## Data access and confidentiality

Data for this retrospective study were accessed for research purposes between December 2017 and November 2018. During data collection, the authors had access to information that could potentially identify individual participants. Following data extraction, all datasets were de-identified prior to analysis, and no identifiable information was accessed or used thereafter.

## Study design

All patients underwent an initial [68Ga]Ga-PSMA-11 PET/MRI after their PCa diagnosis. Subsequently, patients received 0.5 mg of dutasteride per os once daily for seven days. The use of generic drugs was permitted. All included patients were naïve to both dutasteride and androgen deprivation therapy (ADT). A second [68Ga]Ga-PSMA-11 PET/MRI was performed after this seven-day treatment period. The maximum standardized uptake values (SUVmax) of PSMA-positive lesions were measured and compared between the two PET/MRIs. Inter-reader variability was assessed to evaluate the consistency and robustness of image interpretation between independent readers. Following the second PSMA PET/MRI, an image-guided biopsy was performed to evaluate histological features, ISUP grade group, and immunohistochemical PSMA expression in the specimens.

## [68Ga]Ga-PSMA-11 PET/MRI

All patients were injected with [68Ga]Ga-PSMA-11 (range 75–89 MBq) before both scans. After 60 min a standard prostate PET/MRI exam from pelvis to the upper abdomen was performed on a hybrid scanner (Signa PET/MRI; GE Healthcare), first frame over the pelvis was scanned for 15 min and used for this analysis. Scanner and protocol were used in several former studies at our hospital [7,20,21]. In brief: it is a 3-T MRI system with a time-of-flight (TOF) PET detector ring that is installed between the body and gradient coils. There is a 3-dimensional dual-echo for spoiled gradient-recalled echo sequence (LAVA-flex) for attenuation correction. Additional T2 and diffusion-weighted sequences were done over the pelvis with a pelvic PET emission scan measured over 15 minutes to match the time of the MRI sequences. For the abdominal frame, two minutes per bed position were acquired. PET data were acquired in 3D TOF mode in an axial FOV of 153 mm. The emission data were corrected for attenuation using the MRI body composition and iteratively reconstructed (matrix size of 256 × 256 pixels, 3D TOF ordered subset expectation maximization with 3 iterations and 18 subsets, with point spread function, 4.7 mm full width at half maximum, 1:4:1 weighted axial filtering). Furosemide was injected intravenously 30 min before the [68Ga]Ga-PSMA-11 injection (0.13 mg/kg) to reduce tracer accumulation in the bladder. The total protocol time length was 20 minutes. A dual board-certified radiologist and nuclear medicine physician (IAB, > 15 years of experience) analyzed all images, incorporating both the MRI and the PET information as well as all clinical information. The second read was performed by a nuclear medicine physician DAF (>6 years of experience). A volume of interest was placed over areas of increased uptake on both scans (pre- and post-intervention) and determined up to 2 target lesions, using the PRIMARY scoring system with a score of 3–5 [22]. For each lesion $SUV_{max}$ and $ADC_{min}$ were recorded before and after the one-week dutasteride treatment.

## Template biopsy

Biopsies were performed using the BiopSee PSMA PET/MRI-TRUS fusion biopsy system (MedCom, Darmstadt, Germany), which contains computer software (BiopSee 2.0) and a biplanar TRUS probe for fusion of images and planning of the biopsies [23]. The biopsies were performed as outpatient interventions, under general anesthesia, in lithotomy position by specialized urologists. All biopsy cores were assessed by a specialized uropathologist as well as a second board-certified pathologist in case of PCa.

## Histopathological parameters and Immunohistochemistry

Formalin-fixed, paraffin-embedded (FFPE) prostate core biopsy specimens were evaluated on 2 μm hematoxylin and eosin (H&E)-stained sections and subsequently PSMA-stained. From each PET-positive lesion, three core biopsies containing acinar adenocarcinoma were evaluated. Additionally, one to two random biopsies representing PET-negative non-tumorous prostate tissue were chosen. Staging and grading were done according to the WHO/ISUP/UICC guidelines [24,25]. Immunohistochemical staining for PSMA (DAKO, M3620, clone 3E6, 1:25) was performed as described previously [26]. The PSMA expression intensity was quantified using a four-tiered system (0 = negative, 1+ = weak, 2+ = moderate, 3+ = strong) for cytoplasmic PSMA expression by two board-certified, genitourinary pathologists (JHR, NJR) with 9 and 10 years of experience respectively. Further details on the scoring system and representative examples of staining patterns have been described elsewhere [27]. Furthermore, the amount of PSMA-stained tumorous and non-tumorous prostate glands were quantified in steps of 10%. Slides were digitalized (Nanozoomer NDP digital slide scanner C9600-12) using the Hamamatsu NDP.view 2.8.24 Software.

## Statistical analysis

Only descriptive statistics were used given the small sample size (n = 4). Parameters were displayed as mean, standard deviation and range.

## Results

### Patients

Patient characteristics of all four patients included in this pilot study are displayed in Table 1. Mean patient age at the time of the first scan was 68.8 years (±2.5). PSA value at the time of diagnosis ranged from 3.30 ng/mL to 11.86 ng/mL. The mean PSA was 7.16 (±3.53). Multiparametric MRI showed one suspicious PI-RADS-lesion (4–5) in three out of four patients. There was one uncertain PI-RADS-lesion 3 in patient 1. There were no further low-grade lesions diagnosed. PI-RADS 4 lesions were detected in two patients. One patient had a PI-RADS 3, and another had a PI-RADS 5 lesion. All PI-RADS lesions were defined as target lesions for the subsequent template biopsy. 38, 28, 39 and 36 systematic cores were obtained with three targeted lesions in each patient. In three of four patients an ISUP grade of 3 was detected. One patient was diagnosed with an ISUP grade of 2. Incidental carcinoma ISUP 1 and 2 was found in patient 1 in one core

**Table 1. Patients' characteristics and tumor classification data.**

| Patient | Age | PI-RADS | ISUP | PSA (ng/mL) | T-staging | n PSMA+lesions | n cancerous lesions |
|---------|-----|---------|------|-------------|-----------|----------------|---------------------|
| 1 | 68 | 3 | 3 | 3.30 | pT1c | 1 | 1 |
| 2 | 69 | 5 | 3 | 11.86 | cT2a | 2 | 1 |
| 3 | 72 | 4 | 3 | 6.76 | cT2-3 | 2 | 1 |
| 4 | 66 | 4 | 2 | 6.70 | pT1c | 1 | 1 |

each. One incidental ISUP 1 positive core was detected in patient 4. The T-staging varied from pT1c to cT2-3. There was no positive lymph node or organ metastases found in any of the patients.

## Change of SUVmax

We found at least one PSMA-positive lesion per patient (Table 2). In patient 2 and 3, two PSMA-positive lesions were detected. Overall, six PSMA-positive lesions were detected. The SUVmax ranged from 3.1 to 8.0 in the first PSMA PET/ MRI. After one week of dutasteride treatment the second PSMA PET/MRI showed increased levels of SUVmax in five of six PSMA-positive lesions. Reported SUVmax ranged from 3.3 to 9.7 in the second imaging. The increase in SUVmax ranged between 3.6% and 34.7%. Average increase in SUVmax was 12% in PSMA-positive lesions. Only one PSMA-positive lesion (lesion 3) showed a decrease of SUVmax following stimulation. In the corresponding lesion the SUVmax levels decreased by 0.9 in the second imaging, corresponding to minus 11.3%. Changes of SUVmax pre- and post-stimulation are displayed in Fig 1.

We compared the SUVmax of each lesion with the SUVmax of the background. The results are presented in Table 3. The background SUVmax was measured on the contralateral side in a volume of interest (VOI) of the same size, excluding areas of physiological uptake (such as urine) or pathological tissue. The VOI was measured according to the dimensions of each lesion. Except for lesion 1, a notable contrast was observed between the SUVmax of the lesions and that of the background, indicating good lesion visibility in the [$^{68}$Ga]Ga-PSMA-11 PET/MRI scans. The mean of SUVmax$_1$

**Table 2. SUVmax per lesion in the first and second imaging.**

| Patient | Lesion | SUVmax$_1$ | SUVmax$_2$ | ΔSUVmax | ΔSUVmax (%) |
|---|---|---|---|---|---|
| 1 | 1 | 3.1 | 3.3 | + 0.2 | + 6.4 |
| 2 | 2 | 7.2 | 9.7 | + 2.5 | + 34.7 |
|  | 3 | 8.0 | 7.1 | − 0.9 | − 11.3 |
| 3 | 4 | 7.5 | 8.3 | + 0.8 | + 10.5 |
|  | 5 | 7.0 | 7.3 | + 0.3 | + 4.3 |
| 4 | 6 | 5.5 | 5.7 | + 0.2 | + 3.6 |

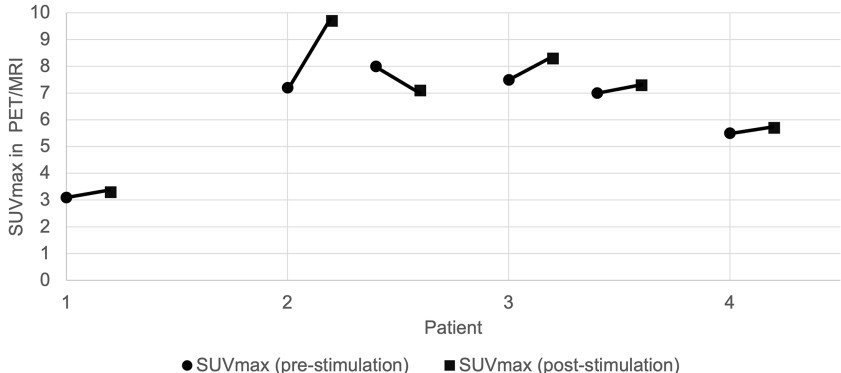

**Fig 1. Changes in SUVmax before and after stimulation across lesions and patients.** Illustrates the SUVmax pre- and post-stimulation in every lesion in each patient. In patients 2 and 3 the first lesion is the malignant one.

**Table 3. SUVmax per lesion and background in the first and second imaging.**

| Patient | Lesion | Lesion SUVmax$_1$ | Background SUVmax$_1$ | Lesion SUVmax$_2$ | Background SUVmax$_2$ |
|---|---|---|---|---|---|
| 1 | 1 | 3.1 | 3.7 | 3.3 | 2.7 |
| 2 | 2 | 7.2 | 3.2 | 9.7 | 3.2 |
| | 3 | 8.0 | 2.5 | 7.1 | 2.7 |
| 3 | 4 | 7.5 | 2.4 | 8.3 | 2.1 |
| | 5 | 7.0 | 2.9 | 7.3 | 3 |
| 4 | 6 | 5.5 | 3.2 | 5.7 | 3 |

**Table 4. SUVmax per lesion by Reader 1 (R1) and Reader 2 (R2).**

| Patient | Lesion | R1 SUVmax$_1$ | R2 SUVmax$_1$ | R1 SUVmax$_2$ | R2 SUVmax$_2$ |
|---|---|---|---|---|---|
| 1 | 1 | 3.1 | 2.9 | 3.3 | 3.3 |
| 2 | 2 | 7.2 | 7.2 | 9.7 | 9.7 |
| | 3 | 8.0 | 8.0 | 7.1 | 7.2 |
| 3 | 4 | 7.5 | 7.5 | 8.3 | 8.3 |
| | 5 | 7.0 | 7.0 | 7.3 | 7.3 |
| 4 | 6 | 5.5 | 5.5 | 5.7 | 5.7 |

was 6.38 (±1.81) vs. SUVmax$_2$ 6.9 (±2.21). This indicates an increase of 8%. The mean background SUVmax$_1$ was 2.98 (±0.49) compared to 2.78 (±0.39) in the second background SUVmax$_2$. This results in a loss of SUVmax of 7%.

The inter-reader agreement is displayed in Table 4, showing nearly identical values of SUVmax between Reader 1 and Reader 2. The overall inter-reader agreement for the two readers was excellent with a Pearson correlation coefficient of 0.99 with almost perfect agreement on the SUVmax.

## Biopsy results

Distribution of PCa, ISUP grade as well as the length of PCa in the core biopsies per lesion and patient are displayed in Table 5 and illustrated in Fig 2C. Prostate biopsies performed after the second PSMA PET/MRI confirmed PCa in four out of six PSMA-positive lesions (lesion 1, 2, 4, 6). Histopathological analysis documented ISUP grade 3 in three out of four patients (patient 1, 2 and 3) in the PSMA-positive lesions. Patient 4 had a maximum ISUP grade 2 in the first PSMA-positive lesion. Less aggressive ISUP grades were detected in all patients. There was only one cancerous biopsy core outside the PSMA-positive lesions showing ISUP grade 1 in patient number 1. In patients 2 and 3 with two lesions no

**Table 5. ISUP of PSMA positive lesions with maximum length of tumor in mm.**

| Patient | Lesion | n ISUP 1 (mm) | n ISUP 2 (mm) | n ISUP 3 (mm) |
|---|---|---|---|---|
| 1 | 1 | 1 (1) | 1 (3) | 4 (2-3) |
| 2 | 2 | 3 (1-4) | 2 (4-5) | 1 (5) |
| | 3 | 0 | 0 | 0 |
| 3 | 4 | 3 (2) | 1 (4) | 2 (3-4) |
| | 5 | 0 | 0 | 0 |
| 4 | 6 | 1 (1) | 5 (2-5) | 0 |

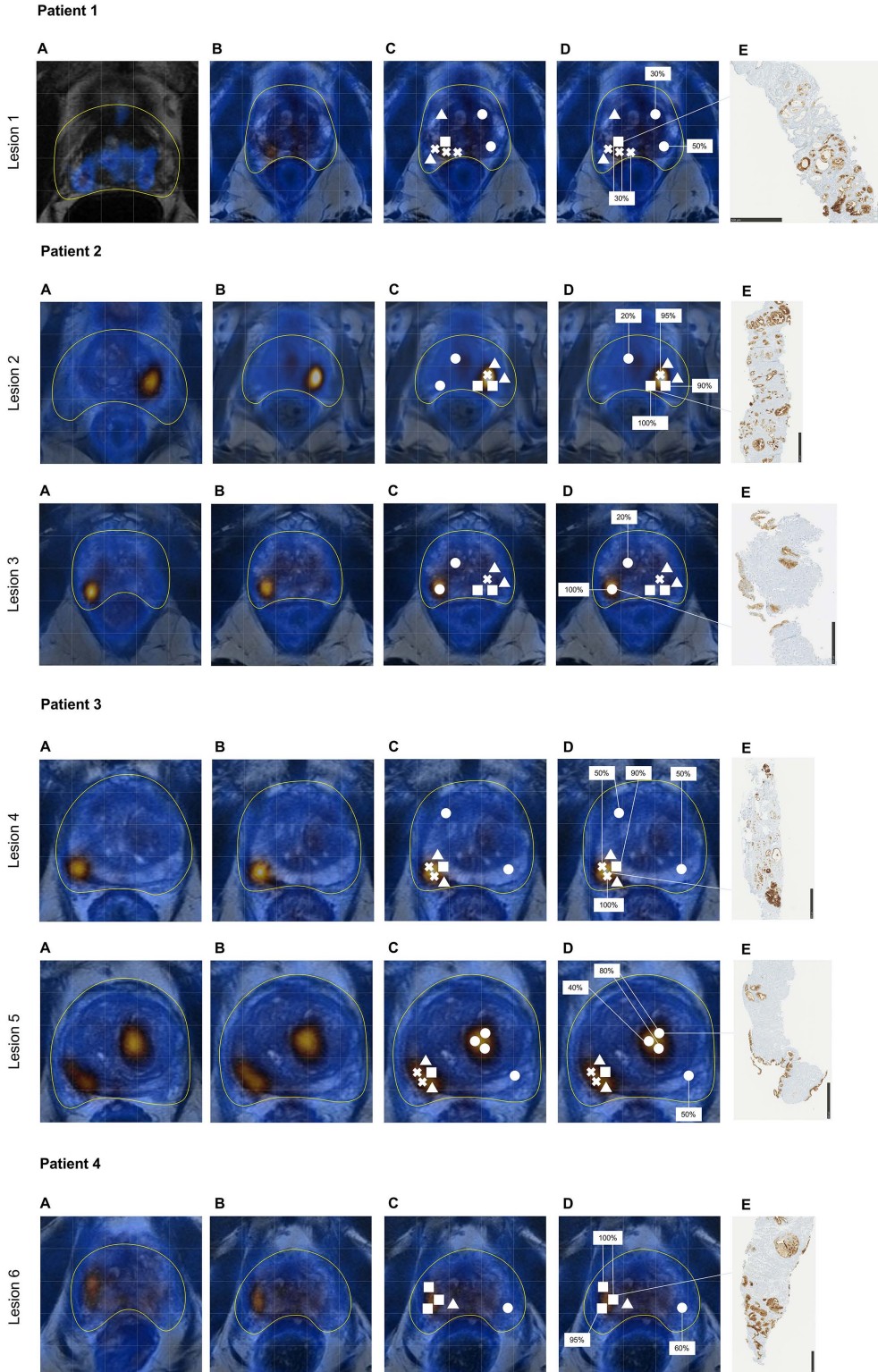

**Fig 2. Sequence of imaging, histopathological, and immunohistochemical analysis of prostate lesions.** Sequence of analysis shown exemplary for all lesions. From top to bottom: Patient 1 to 4: A – Prestimulation PSMA PET/MRI; B – Poststimulation PSMA PET/MRI; C – Biopsy cores: ○ = Benign

tissue, △=ISUP 1, □=ISUP 2, X=ISUP 3; D – Immunohistochemical PSMA expression analyzed by Uropathologists in percentage; E –Histological slide of core biopsy with PSMA staining..

malignant tissue was found in the second PSMA-positive lesions respectively. These results are in concordance with the biopsies, performed prior to the first PSMA PET scans.

### Immunohistochemical PSMA expression

An additional immunohistochemical analysis of PSMA expression was performed (see Fig 2D). In patient 1 and lesion 1, an immunohistochemical PSMA expression of 30% was observed. Control biopsy cores from benign prostate tissue outside the PSMA-positive lesion showed PSMA expression between 30% and 50%.

In patient 2 an immunohistochemical PSMA expression of 90%, 95% and 100% was detected in the biopsy cores from lesion 2. Two biopsy cores with an ISUP grade 2 showed immunohistochemical PSMA expression levels of 100% and 90%, respectively. The biopsy core with an ISUP grade 3 showed PSMA expression levels of 95%. Biopsies performed in lesion 3 showed benign prostatic tissue with a simultaneous PSMA expression of 100%. A random biopsy outside the PSMA-positive region showed a PSMA expression of 20%.

In patient 3 (lesion 4) immunohistochemical PSMA expressions of 50%, 90% and 100% were detected, respectively. Core biopsies with an ISUP grade 3 presented a PSMA expression of 50% and 100%, respectively. The biopsy core with an ISUP grade 2 showed a PSMA expression of 90%. The immunohistochemical PSMA expression in biopsies from lesion 5 showed an expression between 40% and 80%. Control biopsies taken outside the PSMA-positive areas showed a PSMA immunohistochemical PSMA expression of 50% in two biopsy cores.

In patient 4 lesion 6 the immunohistochemical PSMA expression ranged between 95% and 100% and showed a maximum ISUP grade 2. Control biopsies showed an immunohistochemical PSMA expression of 60%.

### Discussion

In this first in human study, short-term treatment with dutasteride was associated with an increase in SUVmax in the majority of PSMA-positive lesions in patients with localized PCa. While the absolute magnitude of change varied between lesions, five of six demonstrated higher values of SUVmax following dutasteride exposure. The largest increase in SUVmax was observed in a high-grade tumor (ISUP 3), suggesting that tumors with more aggressive biological characteristics may exhibit a stronger induction of PSMA expression. This correlation is well-established and reflects the link between tumor aggressiveness, PSA levels, and high PSMA expression levels, which range from 90% to 100% [3,4].

Although the present findings are exploratory and derived from a small cohort, they raise the possibility that short-term dutasteride administration could enhance PSMA expression and thereby potentially improve the detectability of biologically aggressive prostate cancer on PSMA PET imaging. Further studies in larger cohorts are required to confirm this hypothesis and to clarify the clinical relevance of this effect.

Lesion 1 showed a lower PSMA expression compared to benign tissue in the same prostate. This suggests a PSMA-negative cancer in a prostate with highly variable PSMA expression. While PSMA often correlates with cancer aggressiveness, it can be highly heterogeneous, with PSMA-negative areas present in up to 40% of PCa [28,29]. Despite this discrepancy, dutasteride treatment increased SUVmax in this lesion with less pronounced effect on the normal prostatic tissue. The use of dutasteride as a potential pharmacological enhancer of PSMA may have important limitations in patients with predominantly PSMA-negative or highly heterogeneous prostate cancer. In such cases, the possible change in SUVmax due to dutasteride may be more unpredictable, which should be considered when interpreting imaging results and when selecting patients for this approach.

Only one of the PSMA-positive lesions (lesion 3) exhibited a decrease in SUVmax after dutasteride treatment. Histopathological analysis confirmed the presence of benign prostate tissue, while immunohistochemical analysis revealed a heterogeneous pattern of PSMA expression within the tissue. Notably, 100% PSMA expression was observed in this lesion, explaining the positive appearance of the lesion on the PET scan. PSMA expression and uptake are not exclusive to malignant prostate tissue. PSMA is physiologically expressed in normal prostate epithelium, as well as in a variety of non-prostatic tissues (e.g., salivary glands, kidneys, small intestine, and certain neural tissues) and in non-malignant conditions such as benign prostatic hyperplasia, inflammation, and neovascularization [27,30]. PSMA PET imaging can therefore show uptake in benign and non-prostatic pathologies, which is a recognized source of false-positive findings in clinical practice [31–33]. Mechanistically, PSMA functions as a folate hydrolase, with its expression regulated by androgen receptor (AR) signaling among others [34–36]. Dutasteride can upregulate PSMA expression and uptake in prostate cells via modulation of androgen signaling [36,37]. This effect is not limited to malignant cells. Benign prostate epithelial cells also show increased PSMA expression after dutasteride exposure [17]. We do see a limitation in our pilot study concerning that non-malignant mechanisms of PSMA expression are present including physiological expression in normal and benign tissues, upregulation in inflammation and neovascularization, and modulation by androgen signaling. Dutasteride can increase PSMA uptake in both benign and malignant prostate cells by inhibiting AR signaling, thereby potentially enhancing PSMA PET imaging sensitivity but also increasing the risk of false-positive uptake in non-malignant tissue [31,32,36–38].

Two PSMA-positive lesions (4 and 5) were identified in patient 3. The malignant lesion 4 exhibited a more substantial increase in SUVmax and higher immunohistochemical PSMA expression (100%, 90%, 50%) compared to the non-malignant lesion 5 (80%, 80%, 40%), which displayed a relatively small increase in SUVmax and lower PSMA expression. Both PSMA-positive lesions demonstrated a higher PSMA expression than benign prostatic tissue. We detected an overall increase of SUVmax in the PSMA-positive lesions by 8%. The background SUVmax showed decreased levels of SUVmax by 7% after treatment with dutasteride. Based on these findings we hypothesize that dutasteride may preferentially influence PSMA expression in malignant rather than benign tissue, pending confirmation in larger studies.

Current guidelines recommend using PSMA imaging primarily in men with intermediate unfavorable and high-risk PCa and in cases of biochemical recurrence [39–41]. In these scenarios, the primary goal of PSMA imaging is to differentiate between localized or metastatic disease and between local recurrence and metastatic disease. It is well established that detecting low-volume disease at the time of biochemical recurrence can be challenging [7,8]. Conversely, recent studies have demonstrated improved outcomes with salvage radiotherapy when administered at PSA levels below 0.2 ng/mL [9–11]. This discrepancy between low detection rates and better therapeutic outcomes at low PSA levels underscores the need for larger cohort studies with a control arm to validate potential benefits of Dutasteride and other PSMA-altering drugs as radiological enhancers. Further studies are needed to elucidate if Dutasteride may be a valuable tool in distinguishing between metastatic or recurrent disease and non-specific PSMA uptake.

Prior literature on PSMA PET test–retest repeatability has established that quantitative PSMA PET imaging demonstrates solid reproducibility in both tumor and normal tissues. Pollard et al. evaluated [68Ga]Ga-PSMA-11-HBED-CC PET/CT in men with metastatic prostate cancer, showing a within-subject coefficient of variation (wCV) of approximately 12% for bone lesions and 14% for nodal lesions. These metrics indicate that changes in SUV beyond these thresholds are likely to reflect true biological changes rather than measurement variability. However, it is important to note that most published studies have focused on short-interval repeatability (typically within 2 weeks), and data on longer-term reproducibility or in different clinical scenarios (e.g., post-treatment, low PSA states) remain limited [42]. Even though our SUVmax increases were within the range of test-retest variation we do believe that a longer duration of dutasteride as well as a higher dosage might exert these gains. Previous research has demonstrated a dose- and time-dependent effect of dutasteride on PSMA expression levels [38].

This pilot study evaluated the effect of short-term dutasteride administration on the detection rate of PSMA PET/MRI in humans. Given the small sample size the goal was not to detect clinical recommendations but explore preliminary

findings. Further research is needed to determine whether longer treatment durations or higher dutasteride dosages could enhance SUVmax and improve lesion detectability in clinical PSMA PET/MRI imaging.

## Conclusion

In our exploratory pilot study pre-treatment with dutasteride showed increased levels of SUVmax in PSMA-based imaging in prostate cancer. At the moment no clinical implication can be drawn from this data. Further research is needed to optimize this approach and elucidate the underlying mechanisms to potentially improve the detection rates and guiding PSMA-targeted therapies.

## Author contributions

**Conceptualization:** Benedikt Kranzbühler.

**Data curation:** Benedikt Kranzbühler.

**Formal analysis:** Benedikt Kranzbühler.

**Investigation:** Benedikt Kranzbühler.

**Methodology:** Benedikt Kranzbühler.

**Project administration:** Lucas Praetzel, Benedikt Kranzbühler.

**Resources:** Irene A. Burger, Jan H. Rüschoff, Niels J. Rupp, Daniel Eberli, Benedikt Kranzbühler.

**Supervision:** Daniel Eberli, Benedikt Kranzbühler.

**Validation:** Lucas Praetzel, Benedikt Kranzbühler.

**Visualization:** Lucas Praetzel.

**Writing – original draft:** Lucas Praetzel.

**Writing – review & editing:** Lucas Praetzel.

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
