## [Decision Letter · Decision Letter 0]

3 Mar 2026

PONE-D-26-02133Dutasteride treatment and its effect on standardized uptake values in prostate-specific membrane antigen-PET imaging: A pilot studyPLOS One

Dear Dr. Praetzel,

Thank you for submitting your manuscript to PLOS ONE. After careful consideration, we feel that it has merit but does not fully meet PLOS ONE’s publication criteria as it currently stands. Therefore, we invite you to submit a revised version of the manuscript that addresses the points raised during the review process. Please submit your revised manuscript by Apr 17 2026 11:59PM. If you will need more time than this to complete your revisions, please reply to this message or contact the journal office at plosone@plos.org. Please include the following items when submitting your revised manuscript:

We look forward to receiving your revised manuscript.

Kind regards,

Matteo Bauckneht

Academic Editor

PLOS One

**Journal Requirements:**

2. We note that you have provided an approved amendmend from your ethical committee, dated January 2018. With the revised version of your manuscript, please could you also upload the referenced 2016 document, demonstrating that the study recieved continous ethical approval.

3. We note that your Data Availability Statement is currently as follows:

“All relevant data are within the manuscript and its Supporting Information files.”

5. Please remove your figures from within your manuscript file, leaving only the individual TIFF/EPS image files, uploaded separately. These will be automatically included in the reviewers’ PDF.

Reviewers' comments:

Reviewer's Responses to Questions

**Comments to the Author**

1. Is the manuscript technically sound, and do the data support the conclusions?

Reviewer #1: Yes

Reviewer #2: Partly

Reviewer #3: Yes

2. Has the statistical analysis been performed appropriately and rigorously? 

Reviewer #1: Yes

Reviewer #2: Yes

Reviewer #3: N/A

3. Have the authors made all data underlying the findings in their manuscript fully available?

Reviewer #1: Yes

Reviewer #2: Yes

Reviewer #3: Yes

4. Is the manuscript presented in an intelligible fashion and written in standard English?

Reviewer #1: Yes

Reviewer #2: Yes

Reviewer #3: Yes

5. Review Comments to the Author

Reviewer #1: Despite the small cohort, this is an interesting and well-written study that assesses the impact of short-term Dutasteride on PSMA expression based on PSMA PET/MRI. Here are my comments:

“This type II transmembrane glycoprotein is overexpressed in PCa cells”. Precisely in around 90% due to intrinsic PCa heterogeneity, see PMID 40615466 and 39083067

“Even though modern PSMA PET/CTs have a superior diagnostic accuracy than conventional imaging (4), the detection rate remains limited in patients with low-volume disease (low-grade cancer, low PSA) at the time of recurrence (5-7).

Conversely, salvage radiotherapy is most effective in patients with PSA levels below 0.2 ng/mL (8-10).” Indeed, as per the

EAU guideline, a negative PSMA PET should not delay salvage RT.

“Due to the potential side effects (13) of these compounds, recent studies have investigated the impact of dutasteride on PSMA expression in vitro and in vivo (14, 15)”. We now have initial data already reported in an expert paper, such as Vaz S et al. Influence of androgen deprivation therapy on PSMA expression and PSMA-ligand PET imaging of prostate cancer patients. Eur J Nucl Med Mol Imaging. 2020;47:9–15. 10.1007/s00259-019-04529-8. This can be used to extend the introduction (i.e., discussing the hormone therapy's different impact on HSPC and CRPC), also dealing with potential early use of radioligand therapy in the discussion section (i.e., PMID 39934300)

Please describe which Specialists are involved in the interdisciplinary tumour board at the University Hospital of Zurich.

In the whole paper, use EANM radiopharmaceuticals nomenclature (i.e. 68Ga-PSMA-11 à [68Ga]Ga-PSMA-11)

How did the authors select the administered radiopharmaceutical dose?

In the whole paper, use only one term among “MRI” or “MR”.

Add the initials and years of experience of the dual board-certified radiologist and nuclear medicine

Physician, the specialised uropathologist, as well as the second board-certified pathologist.

Did the authors also consider assessing SUVmean and volumetric PSMA PET parameters?

Describe in a little more detail the statistical section: did the authors assess the distribution of each parameter? 68.8 years (66-72) à i.e., it seems that you used the mean and the range, which is not proper.

PI RADS 3 is not suspicious but uncertain.

Were there any PSMA PET-negative PCa lesions?

Another point of discussion may be the different % increase between PCa lesion and background before and after dutasteride.

Did the authors use a standard VOI? Or was it adapted based on each lesion's dimensions?

In patients 2 and 3 with two lesions, no malignant tissue was found in the second PSMA-positive lesions 3 and 5.” Potentially misleading, please rephrase.

The first lines of the discussion should be reorganised in a more proper form as they are more similar to the results description.

“Despite this discrepancy, dutasteride treatment increased SUVmax in this lesion” I would add, “with less pronounced effect on the normal prostatic tissue”.

Reviewer #2: Dear Authors,

your manuscript is very interesting and shows excellent potential. The paper is well written and reads smoothly; the methods are clearly described and detailed. Although the results are based on only four patients, they are appropriately discussed.

However, I would like to raise a few points. Why did you not consider using, as background, a tissue that is likely less sensitive to the drug and more suitable as a reference standard, such as the salivary glands, rather than the contralateral side of the prostate gland?

In general, I believe that the limitations of the results should be highlighted more clearly.

Best regards

Reviewer #3: The pilot study by Praetzel et al. explores the effect of dutasteride on PSMA PET. Despite the small sample size, which limits the ability to draw definitive conclusions, as the authors clearly state, I find the manuscript to be well structured and methodologically sound. It is an excellent starting point for future studies, given the scientific rationale underlying the article and its potentially significant clinical utility.

6. PLOS authors have the option to publish the peer review history of their article (what does this mean?). If published, this will include your full peer review and any attached files.

Reviewer #1: **Yes:** Riccardo Laudicella

Reviewer #2: No

Reviewer #3: No

---

## [Author Response · Author response to Decision Letter 1]

24 Mar 2026

The Response to Reviewers was provided as a separate document as part of the resubmission.

---

## [Editor Report · Decision Letter 1]

20 Apr 2026

Dutasteride treatment and its effect on standardized uptake values in prostate-specific membrane antigen-PET imaging: A pilot study

PONE-D-26-02133R1

Dear Dr. Praetzel,

We’re pleased to inform you that your manuscript has been judged scientifically suitable for publication and will be formally accepted for publication once it meets all outstanding technical requirements.

Kind regards,

Matteo Bauckneht

Academic Editor

PLOS One
---

## [Editor Report · Acceptance letter]

PONE-D-26-02133R1

PLOS One

Dear Dr. Praetzel,

I'm pleased to inform you that your manuscript has been deemed suitable for publication in PLOS One. Congratulations! Your manuscript is now being handed over to our production team.

Kind regards,

on behalf of

Dr. Matteo Bauckneht

Academic Editor

PLOS One